# Investigation of Oxidation Homogeneity in Asphalt Puck after Simulation of Long-Term Aging (Pressure Aging Vessel)

**DOI:** 10.3390/ma16113916

**Published:** 2023-05-23

**Authors:** Lorris Bruneau, Séverine Tisse, Laurent Michon, Pascal Cardinael

**Affiliations:** 1Esso SAF, Avenue Kennedy, 76330 Port-Jérôme-sur-Seine, France; lorris.bruneau@univ-rouen.fr; 2Univ Rouen Normandie, FR3038, SMS, UR 3233, 76000 Rouen, France; pascal.cardinael@univ-rouen.fr

**Keywords:** asphalt, oxidation, pressure aging vessel, Fourier transform infrared spectroscopy

## Abstract

For decades, it has been known that the creation of oxygenated functional groups, especially carbonyl and sulfoxide, is among the main causes of chemical aging and degradation of asphalt. However, is the oxidation of a bitumen homogeneous? The focus of this paper was to follow the oxidation phenomena through an asphalt puck during a pressure aging vessel (PAV) test. According to the literature, the asphalt oxidation process that leads to the creation of oxygenated functions can be divided into the following successive main steps: the absorption of oxygen in asphalt at the air/asphalt interface, diffusion into the matrix, and reaction with asphalt molecules. To study the PAV oxidation process, the creation of carbonyl and sulfoxide functional groups in three asphalts were investigated after various aging protocols by Fourier transform infrared spectroscopy (FTIR). From these experiments performed on different layers of asphalt puck, it was observed that PAV aging resulted in a nonhomogeneous oxidation level inside the entire matrix. Compared to the upper surface, the lower section exhibited carbonyl and sulfoxide indices 70% and 33% lower, respectively. Moreover, the difference in the oxidation level between the top and bottom surfaces increased when the thickness and viscosity of the asphalt sample increased.

## 1. Introduction

Asphalt is the residue of crude oil distillation and exhibits remarkable waterproof and adhesive properties, which make it suitable for pavement construction. Due to climatic conditions and road traffic, asphalt usually degrades during its service life, leading to an accelerated hardening phenomenon and excessive cracking [1,2]. Many studies have been conducted to identify the causes and consequences of asphalt aging. To simulate short-term and long-term asphalt aging in the laboratory, several methods have been developed [3]. Among these methods, the most commonly used is the pressure aging vessel (PAV) [4]. This technique simulates asphalt conditions after several years of road service. As a consequence of the aging process, asphalt undergoes a decrease in physical performance, which can be linked to chemical changes in asphalt, including the creation of carbonyl and sulfoxide functional groups [5,6,7]. Many studies have been carried out to better understand oxygen diffusion within the asphalt matrix and its reaction with asphalt molecules. In general, three steps describe oxygen’s action in asphalt. The first step is the absorption of oxygen at the air/asphalt interface [8]. The second step is the diffusion of oxygen in the matrix. The last step is the reaction between oxygen and asphalt molecules. Dickinson [9] established an equation to estimate the oxygen solubility in asphalt based on Blokker and Van Hoorn’s data [10]. It was observed that the oxygen solubility in asphalt increases when the temperature increases. The oxygen consumed by the oxidation of asphalt molecules was quantified by Herrington [5] using a headspace gas chromatographic method [11]. The results revealed that the rate of oxygen consumption depended on the asphalt composition.

Several studies have reported on oxygen diffusion in asphalt [12,13,14,15,16,17,18,19]. In most studies, Fick’s laws and the Arrhenius equation were used to evaluate the rate of oxygen diffusion [12,13]. Han et al. [14] and Gao et al. [15] studied oxygen diffusion using experimental data and molecular dynamic simulations, respectively. In both cases, oxygen diffusion coefficients were calculated ranging from 10^−10^ to 10^−11^ m^2^/s. The coefficient of oxygen diffusion was influenced by temperature and humidity [16], but also by the binder origin [17]. Other authors highlighted the influence of asphalt hardening on oxygen diffusion during aging [18]. The influence of air voids in asphalt mixtures was also studied by Chen et al. [20], and they proved that the diffusion of oxygen in asphalt mixtures was closely related to the aging susceptibility of asphalt binders.

Most researchers have studied the aging process by following the creation of carbonyl and sulfoxide functional groups during aging, as they can be easily observed using Fourier transform infrared spectroscopy (FTIR) [21,22,23]. Some authors have noticed that the distribution of oxygenated compounds is not homogeneous after aging at atmospheric pressure, with a slightly higher concentration occurring at the air/asphalt interface [5,14,24]. Moreover, Liu et al. [25] highlighted that the aging resistance of asphalt could be increased by reducing the content of several molecules, including molecules containing sulfur and cata-condensed polycyclic aromatic hydrocarbons.

Cui et al. [18] observed that asphalt aging was controlled by the balance between oxygen diffusion inside the matrix and oxidative reactions. The asphalt activation energy used in the calculation of the oxidative reaction rate controls the choice of the predominant step. When this value is low, oxygen diffusion is predominant, and vice versa.

Recently, an equation considering multiple physical fields, including those mentioned above, was developed to accurately predict the degree of oxidative aging across pavement depth for different climate zones [26]. Moreover, our previous work based on statistical analytical calculation demonstrated that different oxidation pathways could coexist in asphalt, depending on the asphalt’s initial composition [27].

Most research on oxygen diffusion and its reaction with asphalt was conducted at atmospheric pressure. It would be interesting to investigate the oxidation process under pressure aging vessel conditions (higher pressure), which is one of the most common aging methods used in the asphalt industry [4]. To better assess the impact of temperature and high pressure on asphalt during a PAV test, in addition to the influence of sample thickness and asphalt nature, an evaluation of the distribution of oxidized compounds inside the asphalt matrix was performed. In addition, the potential diffusion or sedimentation of oxidized compounds in the asphalt matrix was studied. For this study, three asphalts produced from different crude oils with distinct physical properties and chemical compositions were chosen. A modified PAV method was used to simulate asphalt aging. Carbonyl and sulfoxide quantification was performed using infrared spectroscopy. Several experimental protocols were developed to investigate the absorption, diffusion, and reaction of oxygen in asphalt, to calculate the average FTIR indices according to asphalt puck thicknesses, to evaluate the distribution of oxidized compounds in the different asphalt layers, and to study the diffusion of oxidized compounds.

## 2. Materials and Methods

### 2.1. Asphalt

Three asphalts produced by direct distillation of petroleum crudes, namely, F, G, and S, were selected for this study. According to the EN 12591 standard [28], asphalts F and G are within the 70/100 penetration grade, and asphalt F is a 100/150 penetration grade. Their basic properties measured with normalized methods are compiled in Table 1.

### 2.2. Laboratory Aging

Sample aging was carried out using the long-term aging method, the pressure aging vessel (PAV). Usually, PAV conditions are set to 100 °C and 21 bar for 20 h for a 50 g sample (EN 14769 standard [4]). For this study, the conditions reported above were used, except for the aging time and the sample size. For the aging time, the duration was fixed at 24 h. A cup with a 50 mm diameter was used instead of the standard PAV sample pan. The cups were aluminum to allow samples to be demolded (Figure 1). Different sample thicknesses were aged by modifying the quantity of sample asphalt poured into the cup (between 4 and 27 g of asphalt). The conventional degassing step at the end of PAV aging was not performed in order to prevent the oxidized molecules from moving.

### 2.3. Infrared Spectroscopy

During the asphalt aging process, the following oxygenated functional groups are created: carbonyl and sulfoxide. Fourier transform infrared spectroscopy (FTIR) was used to follow the evolution of these specific functional groups during oxidation. For this study, spectra were acquired with a Perkin-Elmer Spectrum Two Instrument using a resolution of 4 cm^−1^ and 16 scans. Attenuated total reflectance (ATR) with a diamond crystal was used to perform the analysis of the asphalt sample. Carbonyl and sulfoxide elongation bands correspond to the areas from 1645 to 1800 cm^−1^ and 970 to 1070 cm^−1^, respectively. Through those two bands, the carbonyl index (CI) (1) and sulfoxide index (SI) (2) were calculated, respectively. In those equations, the areas calculated with the tangential method, corresponding to those two bands, were divided by the area of the alkyl functional group (peak located at approximately 1460 cm^−1^), which is not impacted by aging [29].
(1)$$CI=\frac{Carbonyl\;peak\;around\;1700\;{cm}^{-1}}{Reference\;peak\;around\;1460\;{cm}^{-1}}$$
(2)$$SI=\frac{Sulfoxide\;peak\;around\;1030\;{cm}^{-1}}{Reference\;peak\;around\;1460\;{cm}^{-1}}$$

### 2.4. Experimental Protocol

For each asphalt type, cups with different asphalt masses were prepared. Since the size of the cup was fixed, variations in asphalt thickness were obtained by playing on the volume of the asphalt sample poured into the cup. For the rest of this article, the different sample volumes studied were expressed by sample thickness. The quantity of asphalt poured into the cup varied, reaching asphalt thicknesses between approximately 2 mm (≈4.5 g) and 12 mm (≈26.5 g). For this study, eight different protocols were applied (Figure 2).

In protocols 2 to 8, a freezing step was performed after aging to immobilize oxidized compounds.

When the puck surfaces were being analyzed by FTIR, a small quantity of the sample was collected at both puck surfaces using a hot spatula and placed on the ATR crystal.

When the average asphalt puck indices were being determined, asphalt pucks were homogenized using a spatula.

For this study, a series of protocols was developed to gradually evaluate how the oxidation of bitumen occurs during aging. The first approach (Protocol 1) involved determining whether oxidation could be observed visually. In the second approach, this oxidation was quantified. As a first step, Protocol 2 was proposed to determine the average value of oxidation in the samples tested, in addition to how the sample volumes influenced the results. Following this, Protocol 3 was developed to define whether an oxidation gradient is formed within the samples, and how that gradient is impacted by the volume of asphalt studied. Based on the results obtained, a finer observation of this gradient was performed for better characterization. For this, Protocol 4 was applied. Based on the lessons learned from protocols 1 to 4, Protocols 5 and 6 were proposed to better characterize whether the presence of oxidized compounds in the bituminous matrix could originate from the migration of these molecules. Protocol 7 was suggested to better determine the oxidation phenomena at the air/bitumen interface. Finally, Protocol 8 sought to establish whether oxygen that initially dissolved in bitumen could influence observations.

## 3. Results

### 3.1. Visual Observations of Air (Oxygen) Incorporation into Asphalt Pucks–Protocol 1

During the different experiments conducted for this study, it was observed that after PAV aging, asphalt pucks swelled when they cooled to ambient temperatures. To assess this behavior, Protocol 1 was applied. During the last step of Protocol 1, pictures of the three samples were taken every six minutes. These pictures are presented in Figure 3. For asphalt S, the shape of the sample did not change during cooling. For asphalt G, the shape of the sample changed as the asphalt swelled, swelling which progressively increased with the cooling time. For asphalt F, the same observation was made as for asphalt G, but the swelling was even more significant with the cooling time.

Due to the 21 bar of air pressure applied during the PAV test performed at 100 °C, a certain quantity of air was dissolved in each asphalt sample. Figure 4 shows the sagittal plane of the three samples F, G, and S after 30 min of cooling at ambient temperature.

During the cooling step, the puck expanded based on both the quantity of air dissolved during the PAV test and the capacity of the sample to release this air. This behavior was linked to the viscosity of each tested asphalt (Table 1); the lower the viscosity, the greater the swelling. In addition to this observation, this experiment showed that a “foaming effect”, like the appearance of a sponge, could be observed. This “foaming effect” was due to the release of dissolved gas when the samples returned to atmospheric pressure. The distribution of bubbles in asphalt pucks proved that air (and its oxygen) could diffuse in the entire volume of the sample and was not limited to the sample surface. The distribution of bubbles seemed to be homogeneous in all asphalt pucks. This indicated that aging (asphalt oxidation) could take place in the entire asphalt volume.

It was also observed that after several hours of sample storage under ambient conditions, the shape of the pucks remained swollen for asphalts F and G.

### 3.2. Impact of Sample Volume on Average Oxidation–Protocol 2

Protocol 2 (Figure 2) was developed to determine the relationship between the creation of oxygenated functional groups (carbonyl and sulfoxide) and the volume of asphalt submitted to PAV aging. According to protocol 2, each sample was tested after a homogenization step using infrared spectroscopy. The average carbonyl index (CI) and sulfoxide index (SI) were calculated according to the equation provided above.

First, it was identified that the creation of both oxygenated functional groups in a thin asphalt puck could be related to a specific chemical property. For the sulfoxide functional groups, a link with the elemental sulfur content measured in asphalt (see Table 1) could be established. In contrast, the creation of carbonyl functional groups could be related to the original maltene content, and this fraction can be preferentially attacked by oxygen to create carbonyl functional groups due to its richness in aromatic slightly polycondensed structures [30] (see Table 1).

Figure 5 and Figure 6 illustrate the relationship between the two FTIR indices and the two chemical properties mentioned above. These graphics were established using thinner asphalt samples (2 mm) because it was assumed that all oxidable sites could react with oxygen. These hypotheses seemed to be correct, as good correlations (R^2^ = 0.9981 and R^2^ = 0.9991) were obtained.

Following these first observations, the evolution of the average CI and SI as a function of the asphalt puck thickness after being subjected to PAV aging is shown in Figure 7a,b.

From Figure 7a, several observations can be made. Regarding the average SI, the following was observed:For asphalt F, which contains the lowest elemental sulfur quantity (1.01% w/w), the average SI was quite low and did not change when the thickness of the asphalt puck increased (almost a flat evolution). This suggests that all oxidizable sulfur atoms have been oxidized regardless of the sample volume.For asphalt G, which contains a higher elemental sulfur content (3.34% w/w) than that of previous asphalt, the average SI decreased when the puck thickness increased. This suggests that the available oxygen concentration was not sufficient to oxidize all oxidable sulfur atoms when the sample thickness increased.For asphalt S, which contains the highest elemental sulfur quantity (4.93% w/w), the same observation was made as for asphalt G. In this case, the decrease in average SI is even quicker than for asphalt G.

From Figure 7b, the following observations were made regarding the average CI evolution:For asphalt F, which is the richest in maltene content (98.3 w%), the average CI was the highest among the three asphalts and decreased when the asphalt puck thickness increased.For asphalt G, which contains an intermediate maltene content (89.8 w%), the average CI was lower than that of asphalt F and decreased with increasing asphalt puck thickness.For asphalt S, which contains the lowest maltene content (84.4 w%), the average CI was the lowest, and a trend similar to that of asphalts F and G was observed when the asphalt puck thickness was increased.

Similar to SI, higher CI values were observed for the thinner samples. The main difference with SI involved asphalt F, which did not present a flat evolution. This could occur because contrary to the sulfur quantity, which is limited, many carbon bonds will likely be attacked by oxygen. In addition, the flat evolution of SI observed for asphalt F combined with the highest slope for CI suggested that the creation of sulfoxide and the creation of carbonyl compete.

The decreasing trend observed in Figure 7a,b may have occurred because the same quantities of carbonyl and sulfoxide functional groups were created during aging and became diluted in a larger volume when the sample was homogenized before FTIR analysis. According to these observations, the available oxygen in asphalt pucks may not be correlated with the quantity of sample (the maximum oxygen solubility may not be reached). Moreover, the results suggested that oxidation mainly occurred at the air/asphalt interface. However, the pictures reported in Figure 3 and Figure 4 show that the sample swelled with air, indicating that oxidation occurred in the entire sample volume. To address this point, Protocol 3 was developed.

### 3.3. Impact of Sample Thicknesses on Oxidation at Top and Bottom Puck Surfaces–Protocol 3

To investigate the hypothesis made in the previous section, Protocol 3 (Figure 2) was conducted to study the oxidation that occurred at the top (air/asphalt interface) and bottom of the asphalt pucks. Only the CI was considered here, as the creation of sulfoxide could be limited by the initial content of elemental sulfur. The results are presented in Figure 8 as a ratio calculated according to (3) below.
(3)$$CI\;ratio=\frac{Carbonyl\;index\;measured\;at\;the\;top\;of\;the\;puck}{Carbonyl\;index\;measured\;at\;the\;bottom\;of\;the\;puck}$$

This CI ratio was developed to evaluate how the air penetrated the sample, depending on its thickness, by comparing CI differences between both sample surfaces. The higher the CI ratio is, the larger the difference in CI between top and bottom. As per Figure 8, it was observed that all CI ratios were above 1 regardless of the tested asphalt. As such, the CI at the top was always superior to the CI at the bottom.

Asphalt F displayed a slight CI ratio increase when the puck thicknesses increased, contrary to asphalts G and S, which presented with considerable increases in their CI ratio. The thicker the asphalt puck was, the greater the difference in CI between top and bottom. Consequently, to obtain homogeneous oxidation through the entire sample volume, it is preferable to use thin samples.

Regardless of the puck thickness, asphalt S always presented a higher CI ratio than that of asphalt G, which was greater than that of asphalt F. These differences in CI ratios between the three asphalts could be related to their chemistry, but also to some physical parameters, such as viscosity. This last factor could play a key role in the diffusion of oxygen in asphalt.

To investigate the influence of the kinematic viscosity of the tested asphalts, the relationship between the CI ratio, the original kinematic viscosity at 100 °C, and puck thickness is presented in Figure 9.

From this figure, it was observed that the greater the kinematic viscosity at 100 °C, the larger the CI ratio was, regardless of the sample thickness. This graph confirmed that the original kinematic viscosity of the tested asphalt influenced oxygen diffusion into the sample. This indicated that oxygen could diffuse more easily into the entire asphalt matrix when the kinematic viscosity was low.

This part of the study confirmed that differences in FTIR indices were observed between the top and bottom of an asphalt puck. A higher oxidation level was always observed at the air/asphalt interface, and the differences between both surfaces increased with asphalt puck thickness and with the original kinematic viscosity. However, the oxidation level in the middle of the asphalt puck remains unknown. 

### 3.4. Study of the Oxidized Molecule Distribution in the Different Layers–Protocol 4

To evaluate how oxidized compounds (carbonyl and sulfoxide functional groups) were distributed through the whole asphalt puck after PAV aging, Protocol 4 was conducted, as described in Figure 2. An 11 mm puck made with asphalt G was cut into four equal parts after PAV aging. Each layer was analyzed by infrared spectroscopy. On each layer, CI and SI values were calculated. The results are summarized in Figure 10.

From Figure 10, it was observed that the upper layer (1) was the richest in oxidized compounds. The next two intermediate layers (2 and 3) were characterized by a reduction of 70% and 33%, respectively, of the CI and SI values compared to those of the upper layer. The lower layer (4) was slightly richer than the intermediate layers in oxidized compounds, but its values of CI and SI remained 61% and 24% less rich, respectively, than those of the upper layer and not very different than the two intermediate layers (which could be due to a catalytic effect induced by the aluminum cup or a sedimentation of oxidized compounds). Globally, it was observed that PAV aging was more intense at the air/asphalt interface, which was certainly due to the regular supply of oxygen during the test, and much less intense below the interface. The absence of an oxidation gradient suggested that the diffusion rate of oxygen inside asphalt is not a limiting factor.

Previous experiments showed that the creation of oxidized compounds was limited, either by the quantity of oxygen dissolved in the asphalt puck (due to a limited air/asphalt interface) or by the quantity of oxidized compounds created at the air/asphalt interface. Moreover, the results in Figure 10 demonstrated that except at the upper surface, which contained the highest amount of oxidized compounds, the other part of the puck presented with a homogeneous concentration of oxidized compounds (containing carbonyl and sulfoxide).

To address what happens below the top surface, the following hypotheses were made:

**Hypothesis** **1.**
*Asphalt oxidation occurred at the air/asphalt interface, followed by sedimentation of the produced heavy oxidized compounds.*


**Hypothesis** **2.**
*Asphalt oxidation occurred at the air/asphalt interface, followed by diffusion of the produced oxidized compounds.*


**Hypothesis** **3.**
*The air/asphalt interface was the limiting factor driving the amount of oxygen dissolved in the asphalt matrix (which reacts with asphalt molecules locally).*


These hypotheses about the oxidation process will be further studied with Protocols 5 and 6.

### 3.5. Investigation of Oxidized Compound Diffusion and Sedimentation–Protocols 5 and 6

To investigate Hypotheses 1 and 2, which involved a potential sedimentation or diffusion of oxidized compounds from the air/asphalt interface to the bottom of the puck, Protocols 5 and 6 (Figure 2) were conducted. Several asphalt pucks (homogenized or not) were aged and then heated at 100 °C under a nitrogen atmosphere for 30 h. For each protocol, a reference sample was taken before the nitrogen heating condition was applied. The temperature was set at 100 °C to work with semiliquid asphalt, enabling motion of oxidized compounds. The samples were heated under a nitrogen atmosphere to prevent new oxygenated functional groups from forming in the asphalt puck. The CI was calculated at both surfaces of each asphalt puck, and the results are presented in Figure 11. Through the FTIR indices obtained for Protocol 5, the occurrence of oxidized compound sedimentation (Hypothesis 1) was investigated, and those obtained for Protocol 6 were used to evaluate the diffusion phenomenon of oxidized compounds (Hypothesis 2).

The results of Protocol 5 (Figure 11) showed that the CI calculated at the top and bottom of samples heated or not heated under a nitrogen atmosphere were the same. This absence of an increase in CI at the bottom during these conditions suggested that the sedimentation phenomenon involving oxidized compounds was absent in semiliquid asphalt. Thus, Hypothesis 1 was not confirmed.

The results of Protocol 6 are presented in Figure 11. First, the CI calculated for the reference sample confirmed previous observations. After PAV aging, the CI was higher at the top, approximately two times higher after 24 h of aging for an 8-mm asphalt G puck (reference sample). For the sample heated at 100 °C under a nitrogen atmosphere, the same difference in CI between both surfaces, as in the reference sample, was observed. These identical results between both samples suggested that the diffusion of oxidized compounds was not the limiting factor; thus, Hypothesis 2 was not confirmed.

To summarize, Protocols 5 and 6 showed that during aging at 100 °C under atmospheric pressure, no diffusion or sedimentation of oxidized compounds from the air/asphalt interface to the bottom of the puck were observed.

### 3.6. Investigating the Influence of the Air/Asphalt Interface–Protocol 7

To investigate the role of the air/asphalt interface (Hypothesis 3), especially the size of the exchange surface, on the quantity of oxygenated functional groups created in asphalt during PAV aging, Protocol 7 was conducted. An aluminum sampling cup with a diameter of 70 mm instead of 50 mm was used, corresponding to air/asphalt interface surface areas of 3848 mm^2^ and 2375 mm^2^, respectively. A comparison of the average CI and SI calculated in asphalt samples with the same mass or thickness but with different air/asphalt interface surface areas is presented in Table 2.

From Table 2, differences in carbonyl and sulfoxide indices were observed for the same quantity of asphalt G prepared in two sampling cups with different diameters (samples 1 and 2). For the same masses of asphalt, CI and SI increased when the surface area of the air/asphalt interface increased. By comparing asphalt samples with the same thicknesses but different air/asphalt interface surface areas (samples 1 and 3), the same CI and SI were observed.

From these observations, it was suggested that the evolution of carbonyl and sulfoxide indices was driven by the ratio of the air/asphalt interface surface area divided by the sample mass. This confirmed that the oxidation reaction mainly occurred at the air/asphalt interface. Regardless of the surface area of the air/asphalt interface, the thinner the asphalt sample, the higher the oxidation indices.

### 3.7. Creation of Oxygenated Functions from Oxygen Initially Present in Asphalt–Protocol 8

To study whether the oxygen content initially contained in asphalt (Table 1) could permit the creation of carbonyl and sulfoxide functional groups during PAV aging, protocol 8 (Figure 2) was conducted. The objective of this experiment was to age an asphalt puck without providing oxygen atoms in the PAV tank. Usually, the pressure in PAV was set at 21 bar using air, but for this study, air was replaced by nitrogen, and a total clean-up was performed before aging. The FTIR results obtained for Protocol 8 are given in Table 3.

From Table 3, it was observed that carbonyl functional groups were only created during PAV aging under 21 bar of air (sample 3). After PAV under nitrogen pressure (sample 2), the CI remained at zero regardless of the studied surface. No carbonyl functional groups were created from oxygen atoms initially available in the asphalt matrix during this specific PAV aging.

Concerning sulfoxide functional groups, a small quantity was measured in unaged asphalt (sample 1). It was observed that SI increased after specific PAV aging under nitrogen pressure (sample 2), with a more intense increase at the top than at the bottom. It was also observed that the creation of sulfoxide under nitrogen pressure was less intense than that under 21 bar of air pressure (sample 3). This observation suggested that a small amount of sulfoxide functional groups could be created from oxygen already present inside the asphalt sample. Nevertheless, the oxygen initially contained in asphalt only provided a small contribution to the creation of oxygenated functional groups during PAV aging. Moreover, this experiment also highlighted that the creation of sulfoxide preferentially occurred over the creation of carbonyl.

## 4. Conclusions

In this paper, the objective was to investigate the oxidation process that occurred under PAV conditions (100 °C, 21 bar of pressure). Asphalt oxidation was studied by conducting eight experimental protocols on three asphalts with different physicochemical properties. For PAV aging, an innovative approach involving an aluminum sampling cup was used, producing unmolded, aged asphalt pucks. Working with asphalt pucks made it possible to analyze both the top and bottom of asphalt samples with infrared spectroscopy.

From these eight experimental protocols, the following observations were made:Air was absorbed inside each asphalt sample.The average sulfoxide and carbonyl indices decreased with increasing asphalt puck thickness.The creation of sulfoxide and carbonyl functional groups in a thin and homogenized asphalt puck could be related to the elemental sulfur content and to the original maltene content, respectively.The difference in the oxidation level between the top and bottom of the asphalt puck increased with asphalt puck thickness and kinematic viscosity at 100 °C.PAV aging resulted in a very oxidized thin layer at the air/asphalt interface and a less homogeneous oxidation level for the rest of the puck.No sedimentation and diffusion of oxidized compounds was observed in semiliquid asphalt.A small quantity of sulfoxide functional groups could be created from oxygen initially dissolved in asphalt.

To conclude, PAV aging does not cause a homogeneous oxidation level in the entire asphalt matrix. From the knowledge acquired in this study, it could be suggested that during PAV aging, the oxygen diffusion rate and the oxidation reaction rate were not limiting factors. The air/asphalt interface was identified as a limiting factor, as most of the oxidation reactions occurred at this upper surface and determined the quantity of oxygen dissolved in the asphalt sample. This study presents for the first time that the oxidation depends not only on sample surface area, but also on sample depth and asphalt viscosity. Better knowledge of oxidation phenomena that occur during asphalt aging will enable the limitation of these phenomena by making use of relevant antioxidants to thereby improve the durability of the asphalt [31].

## Figures and Tables

**Figure 1 materials-16-03916-f001:**
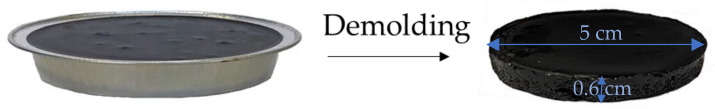
Image of the asphalt cup used for the study before and after unmolding.

**Figure 2 materials-16-03916-f002:**
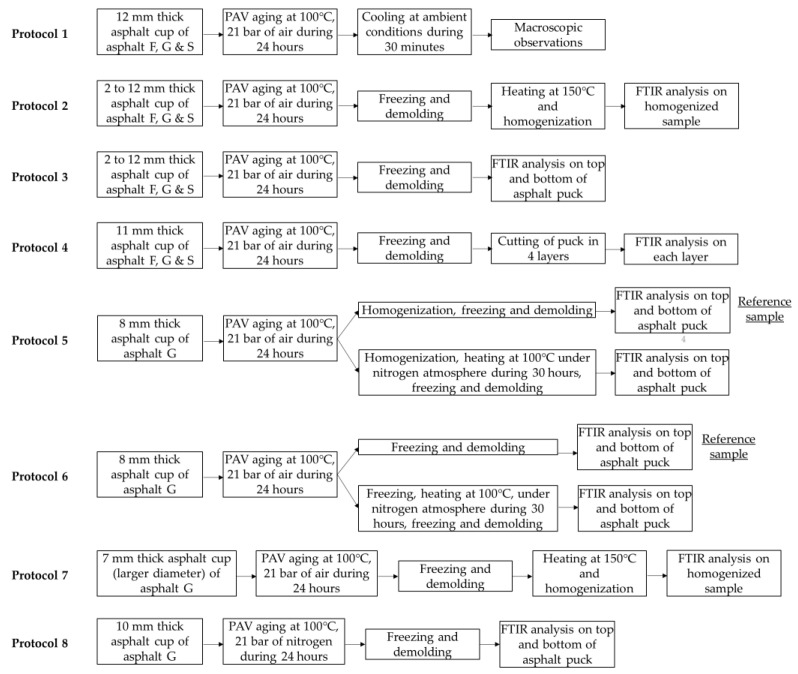
The eight experimental protocols used in the study.

**Figure 3 materials-16-03916-f003:**
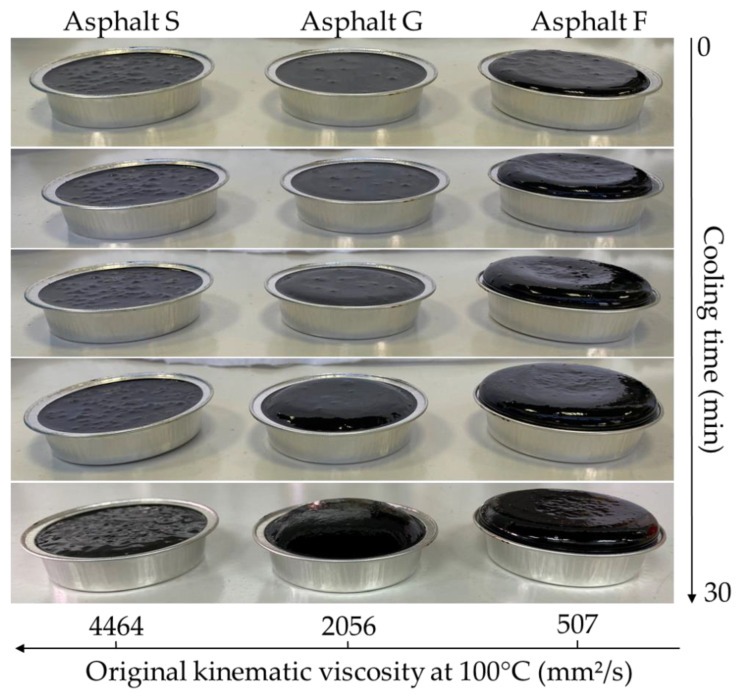
Swelling phenomenon observed for asphalt pucks F, G, and S during cooling at ambient temperature and atmospheric pressure, depending on the kinematic viscosity measured at 100 °C before aging. (Protocol 1).

**Figure 4 materials-16-03916-f004:**
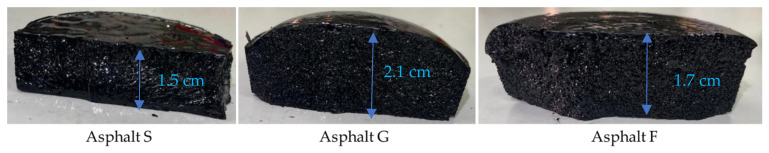
Vertically cut asphalt pucks F, G, and S after PAV aging and 30 min of cooling at ambient temperature and atmospheric pressure. A “foaming effect” was observed. (Protocol 1).

**Figure 5 materials-16-03916-f005:**
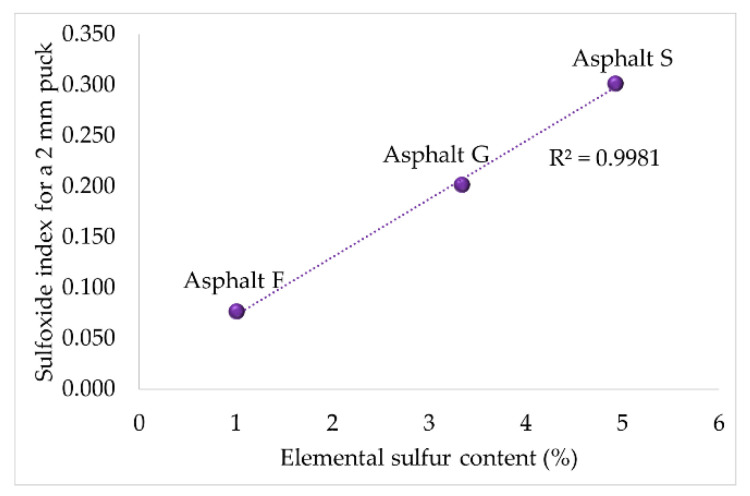
Relationship between the elemental sulfur content and the sulfoxide index calculated for a 2 mm asphalt puck.

**Figure 6 materials-16-03916-f006:**
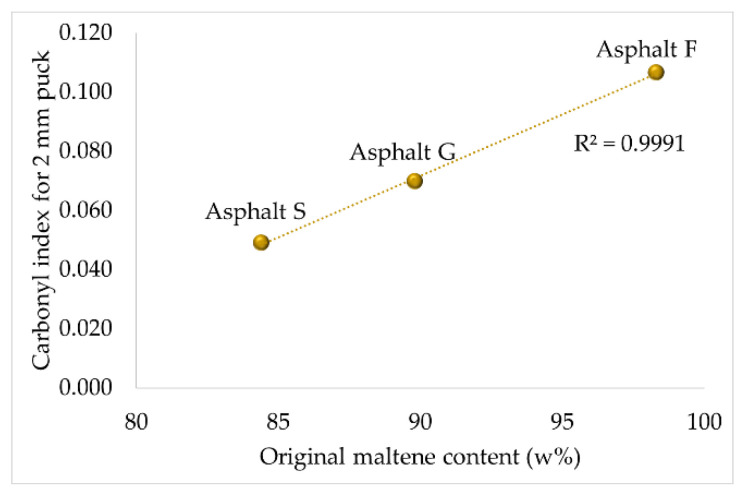
Relationship between the original maltene content and the carbonyl index calculated for a 2 mm asphalt puck.

**Figure 7 materials-16-03916-f007:**
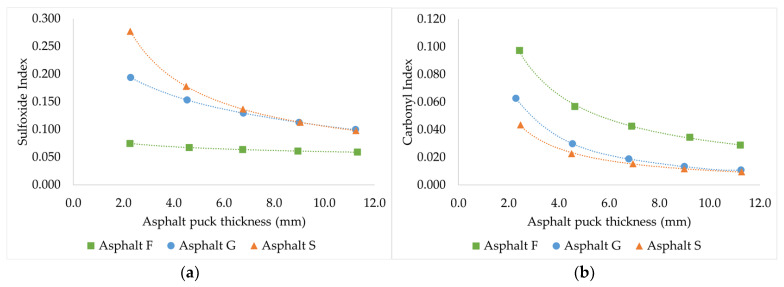
Relationship between the average sulfoxide index ((**a**), on the left) or the average carbonyl index ((**b**), on the right) measured on homogenized sample and the asphalt puck thickness after PAV aging. (Protocol 2).

**Figure 8 materials-16-03916-f008:**
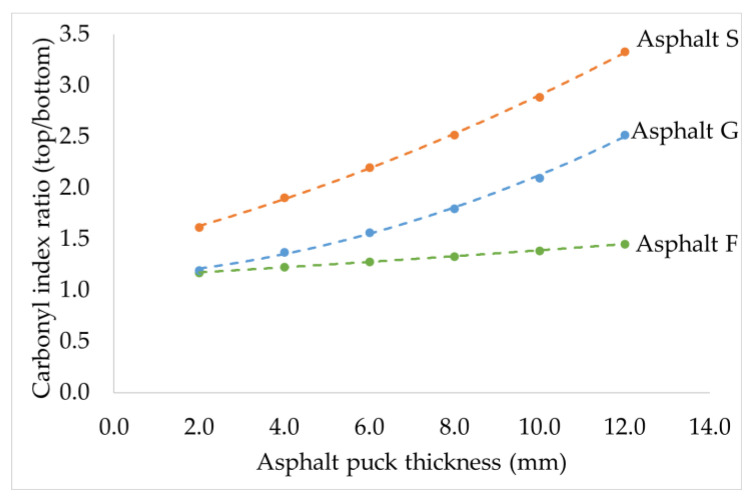
Relationship between the top/bottom CI ratio and asphalt puck thickness for asphalts F, G, and S. (Protocol 3).

**Figure 9 materials-16-03916-f009:**
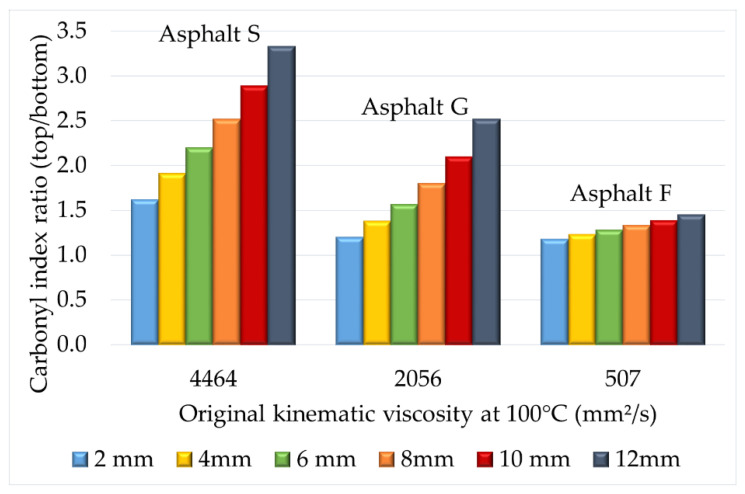
Relationship between the top/bottom CI ratio, kinematic viscosity at 100 °C, and asphalt puck thickness. (Protocol 3).

**Figure 10 materials-16-03916-f010:**
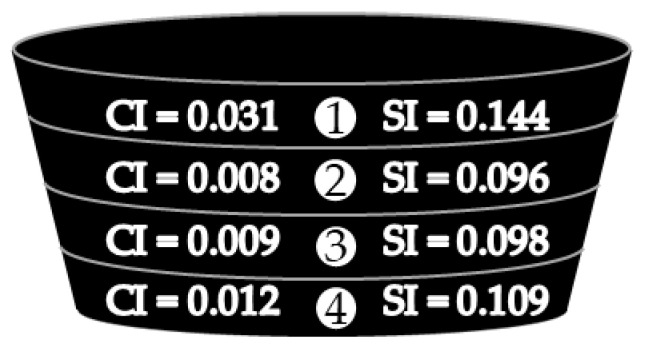
Schematic representation of the carbonyl and sulfoxide index distribution inside the 11 mm asphalt G puck after PAV aging (Protocol 4).

**Figure 11 materials-16-03916-f011:**
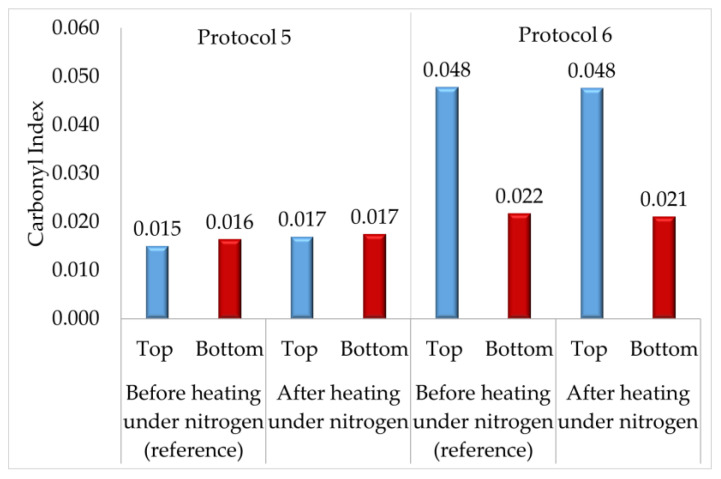
Graphical representation of the carbonyl index evolution at both surfaces of 8 mm asphalt G puck after 30 h at 100 °C under a nitrogen atmosphere. Protocol 5 is presented on the left of Figure 11, and Protocol 6 is presented on the right of Figure 11.

**Table 1 materials-16-03916-t001:** Properties of asphalts F, G, and S.

Test	Method	Unit	Asphalt F	Asphalt G	Asphalt S
Penetration at 25 °C	EN 1426	mm/10	115	77	70
Softening Point	EN 1427	°C	46.2	48.0	48.0
Kinematic Viscosity at 100 °C	EN 12596	mm^2^/s	507	2056	4464
Density	EN 15326	g/cm^3^	0.990	1.027	1.037
Elemental Sulfur Content	MO SR 05-0	%	1.01	3.34	4.93
Elemental Oxygen Content	MA-E2-13	%	0.99	0.93	0.92
Asphaltene Content	NF T60-115	w%	1.7	10.2	15.6
Maltene Content	Deducted from NF T60-115	w%	98.3	89.8	84.4

**Table 2 materials-16-03916-t002:** Evolution of average carbonyl and sulfoxide indices in three asphalt G pucks aged using sampling cups with different diameters. (Protocol 7).

Sample	Mass of Asphalt Poured in Cup (g)	Thickness of Asphalt Puck (mm)	Surface of the Air/Asphalt Interface (mm^2^)	Average Carbonyl Index	Average Sulfoxide Index
1	26.0	7.1	3848	0.019	0.128
2	26.0	11.7	2375	0.010	0.100
3	15.8	7.1	2375	0.021	0.126

**Table 3 materials-16-03916-t003:** Evolution of carbonyl and sulfoxide indices in a 10 mm asphalt G puck after PAV aging under 21 bar of nitrogen or synthetic air. (Protocol 8).

Sample	Aging Conditions	Sample Surface	Carbonyl Index	Sulfoxide Index
1	No aging (reference)	Top	0.000	0.042
		Bottom	0.000	0.041
2	PAV aging (24 h–100 °C, 21 bar of nitrogen)	Top	0.000	0.084
		Bottom	0.000	0.060
3	PAV aging (24 h–100 °C, 21 bar of air)	Top	0.042	0.181
		Bottom	0.020	0.123

## Data Availability

Data and sources available upon request from authors.

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
