# Peer review of "Investigation of Oxidation Homogeneity in Asphalt Puck after Simulation of Long-Term Aging (Pressure Aging Vessel)"

_materials, 2023, doi:10.3390/ma16113916_

Round 1
Reviewer 1 Report
The authors investigate the oxidation process and the physicochemical properties of three asphalts that occurred under PAV conditions. This study could shed light on the oxidation process inside asphalt. The following comments from the reviewer need to be addressed.
1. Line 164, page 5: "the same observation was made as for asphalt G, but the swelling was even more significant with the cooling time", please further explain the reason for different degrees of swelling phenomenon among different asphalts.
2. Line 175, page 6, please provide a detailed definition of the "foaming effect" above.
3. Line 243, page 8: "Similar to SI, higher values were observed for the thinner samples", the meaning of "values" in this paragraph is unknown.
4. Please supplement the frame lines in the upper part of Figure 6 and Figure 7.
5. Line 511, page 15: Please cancel the hyperlink of references.
6. The conclusion part should be improved. Pls detailly state the contribution of this research.
The English writing is professional.
Author Response
Reviewer 1
Point 1: Line 164, page 5: "the same observation was made as for asphalt G, but the swelling was even more significant with the cooling time", please further explain the reason for different degrees of swelling phenomenon among different asphalts.
Response 1: The explanation of the difference of swelling among the three asphalts was detailed Line 165 “During the cooling step, the expansion of the puck grew based on the quantity of air dissolved during the PAV test and on the capacity of the sample to release this air. This behavior was linked to the viscosity of each tested asphalt (Table 1); the lower the viscosity, the greater the swelling.”
Point 2: Line 175, page 6, please provide a detailed definition of the "foaming effect" above.
Reponse 2: Authors agree the comment and the following paragraph was corrected to explain this effect “In addition to this observation, this experiment showed that a “foaming effect”, like the appearance of a sponge, could be observed. This “foaming effect” was due to the release of dissolved gas when the samples was return to atmospheric pressure. The distribution of bubbles in asphalt pucks proved that gas could diffuse in the entire volume of the sample and was not limited to the sample surface”
Point 3: Line 243, page 8: "Similar to SI, higher values were observed for the thinner samples", the meaning of "values" in this paragraph is unknown.
Reponse 3: The correction was added “Similar to SI, higher CI values were observed for the thinner samples.”
Point 4: Please supplement the frame lines in the upper part of Figure 6 and Figure 7.
Reponse 4: The frame lines in the up part was added for Figure 6 and 7.
Point 5: Line 511, page 15: Please cancel the hyperlink of references.
Reponse 5 Two hyperlinks of references were deleted.
Point 6: The conclusion part should be improved. Pls detailly state the contribution of this research.
Reponse 6: The following sentences were added to detail the contribution of this research “This study presents for the first time that the oxidation depends of not only sample surface but also sample depth and asphalt viscosity. Better knowledge of oxidation phenomena during aging through the thickness of the asphalt will allow the limitation of these phenomena by using the most relevant antioxidants and then improve the durability of the asphalt.”

Reviewer 2 Report
Dear Editor of Materials
The manuscript entitled "Investigation of oxidation homogeneity in asphalt puck after simulation of long-term aging (pressure aging vessel)” is an interesting study, the oxidation process that occurred under PAV conditions was investigated by conducting eight experimental protocols on three asphalts with different physicochemical properties. However, in my opinion, the manuscript has some shortcomings in the text. According to the mentioned items, I recommend a minor revision of the manuscript.
Comments for authors
1. The methodology as well as the objective of the research should be provided in the abstract section.
2. Abbreviations should be fully defined on the first use. For example, what is FTIR? It should be addressed.
3. Why these penetration grades (70/100 and 100/150) were applied in this research? Couldn't the authors use a performance grade of 60/70?
4. The index of the figures should be presented below them, not inside the figure.
5. What is the research gap as well as the innovation of this research?
6. More literature should be presented in the introduction section.
7. The authors should present more future directions in the conclusion section. They can recommend various statistical analyses, modeling methods and optimization algorithms to be incorporated into the proposed approaches for further investigation: doi.org/10.5267/j.jfs.2022.11.007, doi.org/10.6007/IJARBSS/v4-i7/1002, doi.org/10.5267/j.jfs.2024.1.004.
8. Try to polish the writing and grammar errors of the manuscript. I found several errors.
9. Enrich the quality of the figures in this research (e.g., Figure 1).
10. Authors should use a third party to write the text. It is better not to use words like "we" (e.g., in line 352).
11. The caption of the figure should be provided next to it. In this regard, correct Figure 11.

1. Try to polish the writing and grammar errors of the manuscript. I found several errors.
2. Authors should use a third party to write the text. It is better not to use words like "we" (e.g., in line 352).
Author Response
Point 1: The methodology as well as the objective of the research should be provided in the abstract section.
Response 1: The objective was specified by adding the following sentence : “ For decades, it has been known that the creation of oxygenated functional groups, especially carbonyl and sulfoxide, is among the main causes of chemical aging and degradation of asphalt but is the oxidation of a bitumen homogeneous? The focus of this paper was to follow the oxidation phenomena through an asphalt puck during a pressure aging vessel (PAV) test.”
The limit of authorized number of words was reached.
Point 2 : Abbreviations should be fully defined on the first use. For example, what is FTIR? It should be addressed.
Response 2: To fully defined FTIR abbreviation, “Fourier transform infrared spectroscopy (FTIR)” was added both in the abstract and in the introduction.
Point 3: Why these penetration grades (70/100 and 100/150) were applied in this research? Couldn't the authors use a performance grade of 60/70?
Response 3: For this research, authors have selected 3 asphalts with significant differences of kinematic viscosity. Asphalt F, G and S were three asphalts available in the laboratory stock with differences of physico-chemical properties. We could also have used a 60/70 penetration grade if this asphalt exhibited desired properties.
Point 4: The index of the figures should be presented below them, not inside the figure.
Response 4: Index inside figures 7 and 11 were removed and presented below them, in the legend.
Point 5: What is the research gap as well as the innovation of this research?
Response 5: The following sentence was added in conclusion part : “This study presents for the first time that the oxidation depends of not only sample surface but also sample depth and asphalt viscosity.”
Point 6: More literature should be presented in the introduction section.
Response 6: Following the advice of the reviewer,,four articles were added in the introduction section:
- Chen, X., Wang, Y., Wen, Y., Cheng, H., & Hao, G., Oxygen Diffusion Coefficients of Asphalt Mixtures and Their Impacts on Mixture Aging. Trans. Res. Rec. 2023, 0(0). https://doi.org/10.1177/03611981221143117
- Das, P.K., Balieu, R., Kringos, N. et al. , On the oxidative ageing mechanism and its effect on asphalt mixtures morphology, Mater. Struct. 2015, 48, 3113–3127. https://doi.org/10.1617/s11527-014-0385-5
- Eman L. Omairey, Fan Gu, Yuqing Zhang, An equation-based multiphysics modelling framework for oxidative ageing of asphalt pavements, J. Clean. Prod. 2021, 280, 1, 124401. https://doi.org/10.1016/j.jclepro.2020.124401
- Shuang Liu, Liyan Shan, Guannan Li, B. Shane Underwood, Cong Qi, Molecular-based asphalt oxidation reaction mechanism and aging resistance optimization strategies based on quantum chemistry, Mater. Des. 2022, 223, 111225. https://doi.org/10.1016/j.matdes.2022.111225.
Point 7 The authors should present more future directions in the conclusion section. They can recommend various statistical analyses, modeling methods and optimization algorithms to be incorporated into the proposed approaches for further investigation: doi.org/10.5267/j.jfs.2022.11.007, doi.org/10.6007/IJARBSS/v4-i7/1002, doi.org/10.5267/j.jfs.2024.1.004.
Response 7: The authors developed statistical analytical calculation on asphalt oxidation and the reference 27 was added in introduction section
“Moreover, our previous work based on statistical analytical calculation demonstrated that different oxidation ways could co-existed in asphalt depending its initial composition [27].”
- Bruneau, L.; Tisse, S.; Michon, L.; Cardinael P. Evaluation of asphalt aging using multivariate analysis applied to Saturates, Aromatics, Resins - Asphaltene Determinator data. ACS Omega, 2023, in press, https://doi.org/10.1021/acsomega.2c07754.
To respond to the remark the following sentence and the corresponding references were added.
“Better knowledge of oxidation phenomena during aging through the thickness of the asphalt will allow the limitation of these phenomena by using the most relevant antioxidants and then improve the durability of the asphalt.”
- Rajabi, M. S.; Habibpour, M.; Bakhtiari, S.; Rad, F. M.; Aghakhani, S. The development of BPR models in smart cities using loop detectors and license plate recognition technologies: A case study. J. Future Sustainability 2023, 3, 75–84. https://doi.org/10.5267/j.jfs.2022.11.007.
Point 8: Try to polish the writing and grammar errors of the manuscript. I found several errors.
Response 8: The article had been corrected by AJE and also by native English speaker (ExxonMobil US colleagues).
Point 9: Enrich the quality of the figures in this research (e.g., Figure 1).
Response 9: The dimensions of the pucks were added on figure 1 and 3.
Point 10: Authors should use a third party to write the text. It is better not to use words like "we" (e.g., in line 352).
Response 10: The authors rewrote the sentence. (“the occurrence of oxidized compound sedimentation (Hypothesis 1) was investigated, and those obtained for Protocol 6 were used to evaluate the diffusion phenomenon of oxidized compounds (Hypothesis 2)”)
Point 11: The caption of the figure should be provided next to it. In this regard, correct Figure 11.
Response 11: The layout around Figure 11 has been revised.
Comments on the Quality of English Language
Point 1: Try to polish the writing and grammar errors of the manuscript. I found several errors.
Response 1: The authors used AJE’s services for English proofreading the certificate was provided to the editor.
Point 2: Authors should use a third party to write the text. It is better not to use words like "we" (e.g., in line 352).
Response 2: The authors rewrote the sentence. (“the occurrence of oxidized compound sedimentation (Hypothesis 1) was investigated, and those obtained for Protocol 6 were used to evaluate the diffusion phenomenon of oxidized compounds (Hypothesis 2)”)

Reviewer 3 Report
I want to congratulate the Authors for a very interesting study on the fundamentals of bitumen aging. I strongly recommend the paper publication after the following minor revisions:
1. Line 49: “-10” and “-11” should be superscripts.
2. Table 1: g/cm3 – “3” should be superscript.
3. Section 2.3: I suggest removing the equipment brand.
4. Section 2.3: how were these areas calculated? With a tangential method or considering the entire area below the spectrum within the reported wavenumber ranges?
5. Figure 7: I suggest using different symbols for the different binders to make the figure readable also in black & white.
Author Response
Point 1: Line 49: “-10” and “-11” should be superscripts.
Response 1: The correction was made.
Point 2: Table 1: g/cm3 – “3” should be superscript.
Response 2: The correction was also made.
Point 3: Section 2.3: I suggest removing the equipment brand.
Response 3: The authors thought that giving the equipment brand is essential to reproduce the data.
Point 4: Section 2.3: how were these areas calculated? With a tangential method or considering the entire area below the spectrum within the reported wavenumber ranges?
Response 4: Before this experiment, authors conducted a study to identify the most accurate method to calculate FTIR areas of an aged asphalt. It was identified that the tangential method was the most accurate and repeatable method. Consequently, we used it to determine all FTIR areas presented in this paper. The following sentence was corrected in section 2.3 to detail the method used “In those equations, the areas calculated with the tangential method and corresponding to those two bands were divided by the area of the alkyl functional group (peak located at approximately 1460 cm-1), which is not impacted by aging [24].”
Point 5: Figure 7: I suggest using different symbols for the different binders to make the figure readable also in black & white.
Response 5: Authors agree the comment and the correction was made in Figure 7a and 7b.
